# *Shigella* Vaccines: The Continuing Unmet Challenge

**DOI:** 10.3390/ijms25084329

**Published:** 2024-04-13

**Authors:** Ti Lu, Sayan Das, Debaki R. Howlader, William D. Picking, Wendy L. Picking

**Affiliations:** 1Department of Veterinary Pathobiology and Bond Life Science Center, University of Missouri, Columbia, MO 65201, USA; drhb7r@missouri.edu (D.R.H.); pickingw@missouri.edu (W.D.P.); 2Department of Microbial Pathogenesis, School of Dentistry, University of Maryland, Baltimore, MD 21201, USA; sdas1@umaryland.edu

**Keywords:** *Shigella* spp., vaccines, shigellosis, type III secretion system, conjugate, live attenuated, whole killed, subunit

## Abstract

Shigellosis is a severe gastrointestinal disease that annually affects approximately 270 million individuals globally. It has particularly high morbidity and mortality in low-income regions; however, it is not confined to these regions and occurs in high-income nations when conditions allow. The ill effects of shigellosis are at their highest in children ages 2 to 5, with survivors often exhibiting impaired growth due to infection-induced malnutrition. The escalating threat of antibiotic resistance further amplifies shigellosis as a serious public health concern. This review explores *Shigella* pathology, with a primary focus on the status of *Shigella* vaccine candidates. These candidates include killed whole-cells, live attenuated organisms, LPS-based, and subunit vaccines. The strengths and weaknesses of each vaccination strategy are considered. The discussion includes potential *Shigella* immunogens, such as LPS, conserved T3SS proteins, outer membrane proteins, diverse animal models used in *Shigella* vaccine research, and innovative vaccine development approaches. Additionally, this review addresses ongoing challenges that necessitate action toward advancing effective *Shigella* prevention and control measures.

## 1. Introduction

Since its discovery in 1897 (Figure 1), *Shigella*, the causative agent of shigellosis (bacillary dysentery), is known to be responsible for causing significant economic and public health burdens on a wide scale. *Shigella* affects over 270 million individuals and causes approximately 212,000 deaths annually in all age groups, constituting about 13% of all diarrhea-related deaths. Among children under 5 years old, *Shigella* is responsible for an estimated 28,000 to 64,000 deaths each year [1]. With a very low infectious dose and emerging antimicrobial resistance, *Shigella* has been categorized as a priority 3 organism on the WHO priority pathogens list [2]. Shigellosis is spread by the fecal–oral route of transmission and enters the environment through improper sanitary measures. The symptoms of shigellosis range from mild watery diarrhea to bloody dysenteric stools with inflammatory intestinal distress, giving rise to fever, abdominal cramps, and scanty stools containing mucus and blood [3].

Collectively, *Shigella flexneri*, S. *sonnei*, S. *dysenteriae*, and *S. boydii* comprise the *Shigella* genus with more than 50 serotypes that continue to become increasingly diverse [4]. With no licensed shigellosis vaccine yet available, the disease continues to persist globally. Recent seminal studies have shed light on shigellosis epidemiology and underscore the need for more effective control measures. Low- and middle-income countries (LMICs) experience elevated endemic infection rates due, in part, to inadequate sanitation and healthcare infrastructure [4]. The effect of shigellosis is most notably seen in children ages 2 to 5, resulting in up to 64,000 deaths annually [5]. Moreover, repeated diarrheal disease in these regions causes impaired growth and development and exacerbates malnourishment in these children [4]. Sporadic outbreaks are also common in refugee camps and military barracks [6]. In 2014, the Global Enteric Multicenter Study (GEMS) conducted extensive research in sub-Saharan Africa and South Asia, employing age-stratified case-control studies for the most prevalent diarrheal diseases. While rotavirus was the most significant cause of moderate to severe diarrhea for children under 1 year of age, *Shigella* became the most prevalent such pathogen for children aged 2 to 5. This study also uncovered significant *Shigella* serotype distribution differences, with *S. flexneri* 2a (SF2a) being the most common serotype encountered; however, *S. flexneri* 6 (SF6) was more frequently seen in the stools of children without diarrhea [7].

An Israeli study explored the dynamics of shigellosis, specifically cyclic epidemics involving *S. sonnei*. This work estimated an average annual incidence of 2425 culture-proven shigellosis cases per 100,000 individuals in Israel between 1998 and 2012. This study noted the protective efficacy of *S. sonnei* infection against re-infection with the same species/serotype (81.8%). Additionally, it showed the importance of comprehensive intervention programs, including the promotion of personal hygiene and potential immunization, to mitigate the transmission and impact of shigellosis in Israel [8]. Another investigation focused on the burden of *S. sonnei*-mediated shigellosis among children (0–59 months) in hyperendemic communities. It reported a cyclic pattern with a mean annual incidence of 10.0 per 1000 in Elad from 2000 to 2013, emphasizing the case-control study’s importance and detailing risk factor analysis methodologies [9]. Furthermore, the latter study employed whole-genome sequencing (WGS) to analyze the microevolution and transmission patterns of *S. sonnei* during these cyclic outbreaks, revealing the division of the Israeli collection into two subclades [10]. Subclade I was prevalent among children from ultraorthodox communities, and Subclade II was more common among children in southern Israel. These subclades highlighted the influence of biogeography and ethnicity on differences observed among children in Israel [10]. In a pivotal 2021 study led by Dani Cohen and colleagues, the safety and immunogenicity of a synthetic carbohydrate conjugate vaccine targeting *S. flexneri* 2a were investigated. This study emphasized the prevalence of the *S. flexneri* 2a (SF2a) serotype in shigellosis cases, particularly affecting the Bedouin population in southern Israel. Notably, it highlighted the protective potential of the SF2a lipopolysaccharide antigen, underscoring its relevance in vaccine development [11]. Cohen’s work also assessed an *S. sonnei*-rEPA (recombinant exopolysaccharide antigen) vaccine for efficacy but found significant pre-existing protection within this group, suggesting prior exposure to a polysaccharide mimic of this antigen [12].

A 2022 study stressed *Shigella*’s global impact, advocating for a WHO-backed vaccine initiative. The authors recommended early engagement with regulators, policymakers, and global health authorities for accelerated licensure and uptake of *Shigella* vaccines, emphasizing scenario planning for regulatory strategies and proactive data collection in late-stage product development to mitigate post-licensure implementation delays [13]. Meanwhile, a Zambia study challenged dysentery as a diagnostic tool for shigellosis in children under 5 years of age, urging improved diagnostic approaches. The findings revealed that relying on clinical dysentery as a screening symptom for *Shigella* infection in children with moderate to severe diarrhea exhibited low sensitivity and a reduced positive predictive value. This raised concerns about the under-diagnosing and under-reporting of *Shigella* infections among children under 5 in Zambia [14].

Likewise, shigellosis poses a significant health challenge in Iran, marked by increasing multidrug-resistant (MDR) strains, particularly involving *S. sonnei*, *S. flexneri*, and *S. boydii* [15]. A recent 9-month study in northeast Iran revealed that over 40% of inflammatory diarrhea cases in children can be attributed to non-duplicate clinical *Shigella* spp. infections, displaying high resistance to commonly used antibiotics like azithromycin, ceftriaxone, and cefixime [16]. Southwest Iran echoes similar concerns, with *S. flexneri* being prevalent, while central regions report surges in cases of both *S. flexneri* and *S. sonnei*, having significant antibiotic resistance to third-generation cephalosporins (ESBL) and ciprofloxacin [17,18]. In contrast, shigellosis research in Afghanistan has focused on outbreaks in military settings, highlighting the risk of reactive arthritis among personnel deployed to developing countries and emphasizing the need for tailored interventions [19]. In the United States, *Shigella* infections exhibited an incidence rate of 9.5 infections per 100,000 child-years, with the highest occurrence in the 1–4 age group (19.5 infections per 100,000 child-years). Furthermore, non-*sonnei Shigella* species were associated with a higher severity rate (30.2%) compared to *S. sonnei* species (13.6%) [20]. *Shigella* spp. have emerged as being among the predominant pathogens associated with acute gastroenteritis outbreaks in schools across the United States [21]. These studies collectively inform our understanding of shigellosis and the need for strategies for mitigation and prevention across serotype boundaries.

Antimicrobial resistance is a growing concern, necessitating innovative approaches for developing effective *Shigella* vaccines. An increase in antimicrobial resistance across multiple nations has become evident from epidemiological studies [22,23,24,25,26]. Several vaccine platforms and target immunogens are being used in an effort to develop a safe and effective *Shigella* vaccine, including whole-killed cell vaccines, live attenuated vaccines (LAVs), LPS-based vaccines, conjugate vaccines, and subunit vaccines [13]. Each approach has distinct advantages but faces inherent hurdles. Despite progress, challenges in coverage, cost, and storage infrastructure still persist, emphasizing the need for ongoing research and technological advancements in *Shigella* vaccine development [27].

This review will attempt to provide a comprehensive overview of *Shigella* pathogenesis and an unbiased assessment of the status of *Shigella* vaccine candidates that are in development. We will highlight the strengths and weaknesses of each vaccination strategy and emphasize the requirements of an effective *Shigella* vaccine. This will include delving into the intricacies and potential obstacles encountered in the use of animal models for research on this human-adapted pathogen. The discussion will include a description of potential *Shigella* immunogens (e.g., outer membrane proteins and conserved T3SS proteins), the use of different animal models, and the development of novel vaccine formulations and approaches. By scrutinizing the existing landscape, we hope to offer insights into the prospects and challenges associated with the development of a *Shigella* vaccine and the likely need for multiple vaccine formulations tailored toward specific target groups.

## 2. *Shigella* Pathogenesis

The genus we know as *Shigella* evolved from *Escherichia coli* ancestors through the loss of certain catabolic pathways and flagellar genes in parallel with the acquisition of a 210–230 kB virulence plasmid (VP), known as pINV (pINV A and pINV B) [28]. Specific examples of the VP include but are not limited to, pMYSH6000 in *S. flexneri* 2a, pSS120 in *S. sonnei*, and pWR100 in *S. flexneri* 5 [29]. *Shigella* is predominantly transmitted through the fecal–oral route, and upon ingestion, they activate acid resistance pathways that include the decarboxylase-independent AR1 pathway and AR2 (the glutamate decarboxylase system). The elevated acid resistance of *Shigella*, which is largely attributed to the glutamate decarboxylate system (*gadA*, *gadB*, and *gadC*), accounts for the low infectious dose of *Shigella* [30]. *Shigella* has a substantially elevated acid resistance relative to many other enteric pathogens, which allows the ingestion of a small number of organisms to result in clinical symptoms [31] (Figure 2). *Shigella’s* access to the colon is facilitated by the ShET-1 and -2 enterotoxins. Once in the colon, it enters the colonic mucosa by initially crossing M cells (Figure 2). Within large intestine lymphoid follicles, resident macrophages take up *Shigella* in a pathogen-driven process, which results in macrophage pyroptosis triggered by the type III secretion system (T3SS) secreted protein IpaB, causing the release of *Shigella* on the basolateral side of the colonic epithelium.

From its position in the lamina propria, *Shigella* invades the overlying colonic epithelial cells by eliciting substantial membrane rearrangements on their basolateral side. Invasion is a function of the mxi-spa operons, which encode the *Shigella* T3SS [32]. Upon host cell contact, the T3SS inserts IpaB and IpaC into the host cell membrane to create the translocon pore, through which later effector proteins are delivered into the host cell cytosol. Among the early effectors is IpaC, which may lead to the activation of Cdc42 and the nucleation of actin at the contact site [33,34]. Other injected effectors, such as IpgB1 and IpgB2, act as guanine nucleotide exchange factors, further reshaping the host cytoskeleton [35]. These and other effector proteins lead to the formation of lamellipodia, filopodia with focal adhesions, and membrane ruffles that enclose the bacterium in a spacious vacuole. The internalized *Shigella* then uses its T3SS to escape into the host cell cytoplasm, where the pathogen replicates. Later effectors, like OspF, then downregulate host immune responses to provide the pathogen with a protected niche for growth. Likewise, OspG interferes with NF-κB activation, thereby further suppressing host inflammation and immunity [36] (Figure 2). A recent study highlights the ability of *Shigella* spp. to induce a “frozen” state in infected cells using IpaJ and VirA to block cellular secretion and trafficking [37]. Once in this replicative niche, *Shigella* utilizes IcsA/VirG for actin-based motility (ABM) in a T3SS-independent process. IcsA is a 120 kDa outer membrane autotransporter protein that is located at a single pole on the surface of the bacterium. In addition to ABM, IcsA can act as an adhesin to contribute to host cell invasion, in addition to mediating cell-to-cell spread [38]. Meanwhile, IcsP (SopA) is an outer membrane protease that cleaves IcsA. IscP has an open reading frame of 981 bp and a deduced amino acid sequence of 327 amino acids [39]. The expression of IcsP is influenced by the transcriptional activators VirF and VirB, and its regulation is distinct from the regulation of IcsA. The tight control of IcsP expression is critical for the modulation of Shigella’s actin-based motility and pathogenicity [40]. IcsB helps the bacteria evade autophagy, a cellular defense mechanism [41]. *Shigella* spp. use those proteins to employ a multifaceted strategy for cellular invasion and evasion of host defenses. Additionally, ShET-1 and -2 enterotoxins contribute to watery diarrhea [42], while Shiga toxin 1 (Stx1) in *Shigella dysenteriae* 1 induces severe post-infection sequelae, including hemorrhagic colitis and hemolytic uremic syndrome [43] (Figure 2). ShET-1 is a virulence determinant encoded by the set1A and set1B genes on the *Shigella* chromosome, and ShET-2 is a 63 kDa protein encoded by the ospD3 gene. Both ShET-1 and ShET-2, as well as Stx1, are proposed to form a holo-AB-type toxin complex in an A1-B5 configuration [44,45].

Understanding the evolution, transmission, and pathogenic mechanisms of *Shigella* spp. is important for providing important information needed for developing effective therapeutic strategies to prevent shigellosis. The complex interaction between *Shigella* and the host underscores the need for targeted interventions to disrupt key virulence factors and enhance the host’s immune responses. Ongoing research continues to unravel the intricacies of *Shigella* pathogenesis, offering hope for the development of novel approaches to combat this significant public health concern [37].

## 3. Challenges Faced in Treating Shigellosis: Antimicrobial Resistance

Shigellosis is typically self-limiting in healthy individuals, but it can be life threatening in malnourished or immunocompromised individuals. The emergence of antimicrobial resistance complicates the public health threat posed by *Shigella*. The recommended treatment regimen to treat shigellosis has changed over time from sulphonamides to ciprofloxacin and azithromycin, and this reflects the changing scope of *Shigella* antibiotic resistance [46,47,48]. Indeed, the United States documented 77,000 cases of antibiotic-resistant *Shigella*, with New York reporting that 20% of isolates demonstrated azithromycin resistance from 2013 to 2015 [49]. Challenges in developing nations, often plagued by poor sanitation and limited outbreak data, have an even more somber outlook [50]. Malnutrition and the overuse of antibiotics further compromise treatment efficacies, particularly in low- and middle-income countries (LMICs) that lack stable healthcare systems and, in many cases, the ability to absorb the economic hardships brought about by illness [51]. Furthermore, studies now show emerging azithromycin, fluoroquinolone, and cephalosporin resistance among *Shigella* clinical isolates [52]. Recognizing fluoroquinolone resistance as a serious public health threat, the World Health Organization (WHO) introduced a separate drug option for fluoroquinolone-resistant *Shigella* infections involving ceftriaxone, pivmecillinum, and azithromycin [53]. Ceftriaxone, a third-generation cephalosporin antibiotic, features a chemical structure comprising a beta-lactam ring fused to a dihydrothiazine ring. Its target genes are the penicillin-binding proteins (PBPs) involved in bacterial cell wall synthesis [54]. Pivmecillinam, a penicillin antibiotic, is structurally based on the penicillin nucleus with a methyl group attached to the amino group. Similar to ceftriaxone, pivmecillinam targets penicillin-binding proteins (PBPs) associated with bacterial cell wall synthesis [55]. Azithromycin, a macrolide antibiotic of the azalide subclass, exhibits a chemical structure comprising a 15-membered lactone ring with various side chains. Its target genes are the 50S ribosomal subunit, particularly the 23S rRNA, which inhibits bacterial protein synthesis [56]. However, resistance to ceftriaxone and azithromycin suggests that the development of extremely drug-resistant *Shigella* (XDR *Shigella*) is imminent [57,58]. Effectively addressing the escalating challenge of drug-resistant *Shigella* strains requires a strategic approach, emphasizing an understanding of resistance dynamics and the exploration of innovative treatments for this global health concern [59].

## 4. The Status of *Shigella* Vaccines in Development

The landscape of *Shigella* vaccine development is rapidly evolving, reflecting the urgency of addressing the significant global burden imposed by *Shigella* infections. The pursuit of a *Shigella* vaccine started with inactivated whole-cell *S. dysenteriae* formulation in the mid-20th century [60]. Over the ensuing decades, numerous vaccine strategies have been explored, from live attenuated strains to subunit vaccines. These efforts have been made possible by a growing understanding of *Shigella*’s complex biology and pathogenicity (Figure 1). The following subsections describe the different types of vaccine strategies that have been explored, which are also summarized in Table 1.

### 4.1. Whole, Killed Cells

Traditional inactivated vaccines are generated through organism killing or inactivation using heat or chemicals. These methods have provided a stalwart vaccine strategy against a wide variety of viral pathogens [61]. Despite their historical importance, they tend to provide inconsistent or weak immune responses that necessitate multiple booster doses for effectiveness [62]. They can also be questioned with regard to being a poorly defined drug substance in the case of bacteria. As a whole, killed *Shigella* vaccine (Figure 3), the *S. sonnei* whole-cell formalin-inactivated vaccine (SsWC) was shown to induce protection against *S. sonnei* in a guinea pig keratoconjunctivitis model, which led to a Phase 1 human trial. Post-vaccination with SsWC, an antibody response against three antigens (SsWC, LPS, and IpaC) was assessed [63]. Although the rise in fecal IgA responses was statistically significant, the increase in IgG levels was not. On the other hand, none of the subjects reported adverse reactions, allowing the authors to conclude that the Phase 1 trial for the vaccine was successful [63]. A potential drawback to SsWC, however, was the fact that it was limited to *S. sonnei* and, therefore, would be expected to provide little benefit to LMICs.

Eventually, the focus shifted to developing multi-serotype vaccines, exemplified by a trivalent vaccine using whole-killed *S. flexneri* 2a, *S. sonnei*, and *S. flexneri* 3a. Varying formalin concentration, temperature, and incubation times during inactivation, different vaccine doses were evaluated in mice (pulmonary infection model) and guinea pigs (keratoconjunctivitis model). Two doses of 10^7^ killed bacteria intranasally induced high anti-Invaplex (IpaB, IpaC, LPS complexes) antibodies in serum and ocular washes. The trivalent vaccine demonstrated protective efficacy against all three strains in the guinea pig model, indicating potential for further development due to its multi-serotype protection and ease of manufacturing [64]. Evaluation of a hexavalent heat-inactivated vaccine covering multiple *Shigella* strains also gave rise to immune responses and protective efficacy in a guinea pig colitis model [65]. Immunized guinea pigs showed protection against individual serogroups with high titers of IgG and IgA against LPS in serum and secretory IgA in intestinal washes. The vaccine’s efficacy was reaffirmed in a rabbit luminal model in a separate study [66].

Killed whole-cell vaccines, while cost-effective and easy to produce, do have inherent drawbacks and risks. They may not be consistently highly immunogenic, thereby requiring multiple boosters. Long-term immunity is not guaranteed. The challenge of serotype cross-protectivity as new clinical isolates emerge remains. Nevertheless, the latter limitation can be addressed by combining multiple serotypes based on regional needs. Unfortunately, reactogenicity has been reported in some cases, and the presence of high levels of LPS makes this a difficult obstacle to overcome. This said, their simplicity, cost-effectiveness, and stability without a strict cold supply chain make them a crucial element in creating an arsenal of vaccine weapons against *Shigella*, especially in LMICs where cost and accessibility are prime considerations.

### 4.2. Shigella Live Attenuated Vaccines (LAVs)

Live attenuated vaccines (LAVs) against *Shigella* (Figure 3), designed for oral administration, aim to induce a safe and effective immune response by eliminating specific genes needed for virulence and/or bacterial survival. The primary targets in *Shigella* LAVs currently in the pipeline include *virG* (*icsA*/*virG* is required for *Shigella* cell-to-cell spread) and *guaBA* (needed for guanine nucleotide biosynthesis). The *S. sonnei* live *virG* knockout vaccine candidate, WRSs1, demonstrated safety, tolerability, and immunogenicity in Bangladeshi adults and children [67]. Deletion of the *virG*/*icsA* gene in WRSs1 generated a strong immune response with higher tolerability and relatively mild reactogenicity. WRSs1 showed promising results in animal models [68], as well as in North American [69] and Israeli volunteers [70]. To address the reactogenicity of WRSs1, *S. sonnei* live attenuated vaccine candidates, WRSs2 and WRSs3, contained additional mutated genes related to ShET-1 and -2 enterotoxins activity and LPS modification. Both strains displayed improved safety and comparable immunogenicity in animal models, with lower reactogenicity [71,72,73]. Phase 1 trials confirmed their safety and immunogenicity over a wide dose range in human volunteers [74].

Other live attenuated candidates in early clinical development include *S. flexneri* 2a SC602, which showed protection in mouse models and North American volunteers [75,76,77,78] but gave rise to a limited IgG response in Bangladeshi volunteers [79]. It could also cause varying degrees of reactogenicity in a dose-dependent manner. A human papillomavirus (HPV) antigen vaccine in which HPV16L1 was expressed by the *Shigella* SC602 vaccine strain was explored to allow for alternative HPV vaccine delivery (conjunctival or intrarectal) in a guinea pig model, though this work was geared more toward viral antigen delivery using *Shigella* as a vector [80]. Similarly, live attenuated *S. dysenteriae* 1 SC599 exhibited good immunogenicity and high tolerance but showed strong serotype specificity [81,82]. This particular *S. dysenteriae* LAV had *icsA*/*virG* deleted along with the iron acquisition genes *ent* and *fep* and the Shiga toxin A subunit (*stxA*).

The *S. dysenteriae* 1 live attenuated vaccine candidate, WRSd1, harbors a deletion of *icsA*/*virG* in addition to the deletion of *stxA* and *stxB* to eliminate both Shiga toxin subunits. This LAV demonstrated protection in the guinea pig keratoconjunctivitis model and proved safe when delivered by the intragastric route in Rhesus monkeys. A combination vaccine containing *S. sonnei* WRSs1, *S. flexneri* 2a SC602, and *S. dysenteriae* WRSd1 also induced protection against homologous challenges [83]. There was, however, lower protection for the WRSd1 strain in the combination when compared to the same strain administered as a single, pure culture. Phase 1 trials showed no fever or shigellosis among volunteers, with most of them showing a strong IgA response to the *S. dysenteriae* LPS following oral delivery of WRSd1 [84]. Recently, researchers found that vaccination of WRSs1 elicited functional antibody responses, including serum bactericidal antibody (SBA) and antigen-specific serum IgG antibodies [85]. SBA titers, which facilitate complement-mediated killing, differed between adults and children, with higher titers observed only in children. Meanwhile, antigen-specific serum IgG antibodies were modestly increased after vaccination with the WRSS1 vaccine in children, suggesting that the younger may be impacted more by the bactericidal antibodies [85]. Moreover, WRSs1 also induced Th1/Th17 signature cytokines in Bangladeshi adults and children [85].

Second-generation live attenuated vaccine candidates, including WRSf2G11, WRSf2G12, and WRSf2G15, showed improved protection and lower reactogenicity by deleting additional genes (e.g., set, *senA*, and *senB* enterotoxins) compared to SC602 [86]. These candidates induced robust immune responses in the guinea pig model with reduced dose-dependent reactogenicity compared to the parent LAV strain [87,88].

The *S. flexneri* 2a live attenuated vaccine candidate, CVD 1207, with deletions in *icsA*/*virG*, *sen*, *set*, and *guaBA*, demonstrated relatively low reactogenicity and good immunogenicity in a dose escalation study [89]. *S. flexneri* CVD 1208S and CVD 1204 (expressing enterotoxigenic *E. coli* (ETEC) colonization factors and detoxified heat-labile enterotoxin or LT) again demonstrated the relative safety of these LAV platforms and the utility for *Shigella* LAV presentation of foreign antigens and CVD 1208, deleting *guaBA* and *Shigella* enterotoxin genes [89,90]. Mutants of *guaBA* were then explored for the presentation of additional ETEC antigens in a *Shigella* LAV strain [91,92]. The combined *Shigella*-ETEC vaccine candidate, CVD 1208S-122, provided further evidence of a *Shigella* LAV’s ability to induce antibody responses and provide protection against diarrhea and weight loss in murine models of *S. flexneri* and ETEC infections [93]. Recently, the live attenuated *S. sonnei* vaccine strain, CVD 1233-SP, has been evaluated in the human enteroid model, demonstrating its ability to invade cells while losing intracellular replication, a crucial attenuating feature, and improved stability through plasmid maintenance system mutations, emphasizing its promise for *Shigella* vaccine development [94].

It is now a growing trend to develop live attenuated combination vaccines targeting different bacterial pathogens simultaneously. ShigETEC, a live attenuated combined vaccine targeting *S. flexneri* 2a and ETEC, exhibited encouraging results in animal models and is progressing through Phase 1 clinical trials [95]. This oral vaccine candidate demonstrated good tolerance over four doses at 3-day intervals, with minor and transient reactogenicity, and indicated potential benefits of a five-time dose regimen for enhanced anti-ETEC immunity based on increased antibody response rates and levels [96]. A novel *Salmonella* vaccine system, employing regulated delayed antigen synthesis (RDAS) and regulated delayed expression of attenuating phenotype (RDEAP) systems for delivering *S. flexneri* 2a (Sf2a) O-antigen, has also been described [97]. Controlled by the LacI-repressible P trc promoter, this platform could be induced to synthesize native Salmonella LPS in the presence of arabinose and triggered Sf2a O-antigen synthesis in the absence of arabinose, demonstrating its potential for effective vaccines against *Shigella* and *Salmonella* infections [97]. A RASV-delivered Sf2a, O-antigen can induce robust CD4+ T-cell responses and IgG responses, thereby warranting further research to develop *Shigella* vaccine candidates using RASV [98]. Furthermore, an oral *Shigella* 2aT32-based vaccine expressing a fusion antigen of UreB-HspA, which were important vaccine candidate antigens for *Helicobacter pylori*, has been evaluated as a potential vaccine against both *Shigella* and *H. pylori* in a mouse model [99].

While oral live attenuated vaccines offer economic advantages for low- and middle-income markets, concerns include the potential recovery of virulence and the need for proper storage conditions, which limit their utility in developing countries. Likewise, there is the potential for dose-dependent symptomology, especially in vulnerable populations (e.g., immunocompromised individuals), and the shedding of LAV strains is always a possibility. Research and new technologies are essential for the ongoing development of LAV *Shigella* strains and antigen delivery systems to address potential problems and enhance their potential benefits in preventing shigellosis in developed nations and in LMICs.

### 4.3. LPS-Based Vaccines

Serotype-specific protection for *Shigella* whole, killed cell vaccines and live attenuated vaccines is a result of surface somatic antigen being the dominant immunogen presented in these vaccine platforms. Because of this, the direct use of LPS as a vaccine component has become an alternative platform for *Shigella* vaccine development. The focus on lipopolysaccharides (LPSs) in *Shigella* vaccine development centers on its three components: O antigen (the outermost and exposed repetitive glycan polymer), core oligosaccharide (conserved LPS core to which the O antigen is anchored), and lipid A (the acylated and phosphorylated di-glucosamine component that anchors LPS to the bacterial outer membrane). The O antigen is a crucial target for the host humoral immune response to whole *Shigella.* Unfortunately, it is responsible for the serotype differences between and within the different *Shigella* species, limiting the ability of whole cell vaccine platforms to offer broad protection. A recent study using the controlled human infection model (CHIM) showed that LPS-specific serum IgA and IgA-secreting memory B cell responses were related to host survival following the *S. sonnei* 53G challenge [100]. Therefore, LPS-associated IgAs are canonically considered an essential component of a successful protective immune response. This mindset, however, is largely based on the protection seen after natural exposure/infection.

An attractive LPS-based vaccine candidate has been named Invaplex, and it consists of *Shigella* LPS combined with the T3SS proteins IpaB and IpaC (Figure 3). A natural version of Invaplex was originally described and is derived from a water extract of virulent *Shigella*, which contains LPS, IpaB, IpaC, and IpaD [101]. A defined version of Invaplex was generated using extracted *Shigella* LPS mixed with purified recombinant IpaB and IpaC [102]. Comparative studies in mice revealed the potency of the artificial Invaplex in inducing antibodies against LPS, with the artificial Invaplex displaying higher serum antibodies against IpaB and IpaC and efficacy at a 10-fold lower dose in the guinea pig keratoconjunctivitis model relative to the natural complex found in water extracts [102]. Invaplex AR, the second-generation artificial Invaplex product, has been found to possess inherent adjuvanticity and a demonstrated safety and tolerability profile. It was also able to induce robust systemic and mucosal immune responses in Phase 1 studies [102]. This vaccine candidate has also undergone a successful cGMP manufacturing feasibility study and can be manufactured to exhibit an expanded serotype coverage [103,104].

Another promising LPS-based vaccine candidate, Ac3-S-LPS, was specifically designed for human use. This tri-acylated lipid A variant induces the production of IgG and IgA antibodies with minimal adverse events, positioning it as a prospective candidate for targeting LPS in preventing human shigellosis [105]. Using a similar approach, the same research group designed a pentavalent LPS candidate vaccine targeting *S. flexneri* 1b, 2a, 3a, 6, and Y (PLVF), which can elicit protective immunity by generating bactericidal antibodies targeting somatic antigen [106]. Other researchers have explored the development and evaluation of needle-free delivery systems for outer membrane vesicles (OMVs) derived from *S. flexneri* as a pseudo-subunit vaccine [107,108].

Despite these advances, LPS encounters significant limitations because it is a T-cell-independent antigen, thus only activating the humoral arm of the immune system. To overcome this limitation, conjugation with carrier proteins becomes essential to induce the cell-mediated immune responses needed to establish robust immunological memory. While LPS alone lacks substantial value as a vaccine candidate, innovative strategies may be able to leverage its benefits in *Shigella* vaccine development, thus addressing some of the current challenges in a rapidly evolving field.

### 4.4. Conjugate Vaccines

As a major antigen involved in the eventual clearance of *Shigella* during a natural infection, the surface somatic antigen (O-antigen) or LPS is an attractive vaccine target, but it is also what gives rise to the serotype diversity within *Shigella*. Nevertheless, several conjugate vaccine candidates are being developed consisting of O-specific polysaccharide (O-SP) conjugates linked to protein carriers, such as succinylated mutant *Pseudomonas aeruginosa* exotoxin A (rEPA) or the *Corynebacterium diphtheriae* toxin mutant CRM9 [109,110,111]. When tested in humans, these conjugates demonstrated a good safety profile and high efficacy. With ongoing formulation improvement, they have now progressed to Phase 3 trials [12,112]. A noteworthy candidate, Flexyn2a [113], was generated by combining *S. flexneri* 2a O-SP and EPA, and it exhibited promising outcomes in Phase 1 trials where it was demonstrated to be safe with good immunogenic potential [114] (Figure 3). Two other successful bioconjugates, Sd1-EPA and Sf2a-EPA, were developed by expressing and assembling *S. dysenteriae* type 1 (Sd1) and *S. flexneri* 2a (Sf2a) O-polysaccharides, respectively, on an *E. coli* glycosyl carrier lipid, which was then conjugated to EPA [115,116]. Combining these, a quadrivalent bioconjugate vaccine (*Shigella* 4V) was generated and was shown to induce functional antibody responses to targeted serotypes. The addition of an aluminum salt adjuvant enhanced the response and cross-reactivity, suggesting potential broader protection than initially anticipated [117].

Similarly, GVXN SD133 is a conjugated *S. dysenteriae* vaccine featuring *Shigella* O1 LPS and EPA (Figure 3). It demonstrated good safety and humoral immunogenicity in Phase 1 trials [118]. Exploring broader protection, conjugates of *S. sonnei* and *S. flexneri* 2a O-SP were identified to induce cross-reactive protective antibodies against both of these *Shigella* strains in human volunteers [119]. An innovative approach involves developing conjugates encompassing adhesins, polysaccharides, and LPS from ETEC, *Campylobacter jejuni* (CJ), and *S. flexneri*, which has demonstrated immunogenicity in mouse models [120]. A recent study in China used the ZF0901 vaccine, comprising detoxified O-specific polysaccharides from LPS of *S. flexneri* 2a and *S. sonnei* covalently linked to tetanus toxoid, and demonstrated it was safe and immunogenic in infants and young children, inducing robust immune responses characterized by significantly elevated levels of type-specific serum antibodies against both *Shigella* serotypes [23]. A multivalent *Shigella* conjugate vaccine (SCV) targeting *S. flexneri* 2a and *S. flexneri* 3a was generated using squaric acid chemistry and found to induce high-level and durable immune responses to the O-specific polysaccharide (OSP) component of lipopolysaccharide when administered intramuscularly [121].

Despite progress, the limits posed by *Shigella* conjugate vaccines lie in narrow serotype and strain coverage, thereby failing to provide a broad protection encompassing all *Shigella* species and strains. The utilization of synthetic oligosaccharide-based vaccine candidates, aiming to address these challenges by delivering artificial *Shigella* oligosaccharides with low toxicity, has shown some promise [122,123,124]. However, while some candidates demonstrated considerable immunogenicity in mouse models, the protection was specific to the spectrum from which the synthetic oligosaccharide was derived. Meanwhile, a new *Shigella* O-polysaccharide (OPS)-IpaB conjugate vaccine, which uses the T3SS protein IpaB as the carrier protein, has been identified that could induce strong antibody responses and provide protection against multiple *Shigella* serotypes by virtue of the IpaB portion, suggesting its potential as a more broadly effective measure against shigellosis [125].

In *Shigella* vaccine development, conjugate vaccines targeting O-Ags represent a viable pathway forward, delivering safe and effective outcomes at moderately low costs. The current strategies, despite exhibiting several advantages, face limitations such as a lack of suitable animal models for in-depth testing and a lack of understanding of the immune mechanisms they trigger. To overcome these challenges, emerging technologies from molecular biology and glycobiology, combined with structural and chemical biology and glycoengineering, especially the synthetic glycol-based strategy, offer new avenues to explore and enhance our understanding of the immunogenicity induced by glycoconjugate vaccine candidates.

### 4.5. Subunit Vaccines

Subunit vaccines (Figure 3), driven by small-scale pathogen-derived protein components, are gaining popularity in advanced vaccine development due to modern molecular biology and recombinant protein technology. Despite their potential, small subunit proteins often lack the necessary immunogenicity for a protective response [126], necessitating a focus on identifying immunogenic targets and immune-enhancing formulation platforms in subunit vaccinology.

#### 4.5.1. Vaccines Targeting the T3SS

Among the primary targets for subunit vaccine development against *Shigella* infections are the conserved type III secretion system (T3SS) proteins IpaD and IpaB. The localization of these two proteins at the tip of the exposed needle of the T3SS injectisome, which is *Shigella*’s most important virulence factor, makes them attractive targets for preventing disease onset. Vaccines based on these proteins have demonstrated strong protection against multiple *Shigella* serotypes. In a mouse lethal pulmonary model, intranasal immunization with IpaB or IpaB with IpaD, together with the double mutant heat-labile toxin (dmLT) of ETEC as a mucosal adjuvant, induced both B cell and T cell immune responses, resulting in full protection against *S. flexneri* and *S. sonnei* [127]. Additionally, IpaB and IpaD demonstrated the potential for inducing systemic or mucosal responses through different administration routes, such as intradermal or sublingual routes [128]. Building upon these findings, a fusion protein consisting of IpaD and IpaB (DB fusion or DBF) was generated and found to elicit immune responses and protection from lethal *S. flexneri* and *S. sonnei* challenge in mouse models when administered intranasally with dmLT as the adjuvant [129].

Enhanced approaches to purifying IpaB-containing recombinant proteins, such as using N,N-dimethyldodecylamine N-oxide (LDAO) for chaperone removal, contributed to enhancing these recombinant proteins’ ability to induce IL-17 expression and protect mice against *Shigella* challenge [130]. Recently, a self-adjuvanting vaccine candidate, L-DBF, was developed by combining LTA1, the active moiety of lethal toxin from ETEC, with DBF [131]. By employing LTA1 as part of the fusion antigen, the risk of Bell’s palsy caused by dmLT could be circumvented [132]; however, the adjuvant potential of dmLT was retained. The L-DBF vaccine demonstrated protective efficacy against multiple *Shigella* spp. serotypes. Moreover, immune responses induced by L-DBF were not adversely affected by prior exposure to *Shigella* spp. [133]. As part of an oil-in-water emulsion formulated with BECC438 (a detoxified lipid A analog known as Bacterial Enzymatic Combinatorial Chemistry candidate #438), L-DBF has a very high protective efficacy against *Shigella* challenge at very low doses, inducing high IFN-γ and IL-17 secretion from mucosal site lymphocytes [134].

Displaying IpaB or IpaD antigens on the surface of bacterium-like particles (BLP) from Lactococcus lactis, known for targeting TLR-2, also leads to robust mucosal adjuvanticity [135]. BLP-IpaB/D induces broad immune responses and produces antibodies with enhanced neutralizing abilities, providing protection for adult mice and assisting newborns in surviving *Shigella* challenge [136]. Moreover, a promising subunit-based vaccine against *Shigella* spp., incorporating IpaB, IpaD, StxB, and the autotransporter IcsA (VirG) into the chimeric protein, was shown to elicit effective antigen-specific immune responses, neutralizing activity against the Shiga toxin, and significant protection against *Shigella* challenge in both in vitro and in vivo experiments [137].

IpaC also demonstrates remarkable immunogenicity and protection against *S. dysenteriae* 1 challenge in guinea pigs [138]. A multi-subunit protein comprising CfaB (ETEC adhesin), IpaC, and intimin (enterohemorrhagic *E. coli* or EHEC adhesin) induces significant immune responses against ETEC, *Shigella*, and EHEC in mouse and guinea pig models [139]. A self-adjuvanting, intranasal recombinant vaccine for shigellosis based on stabilized IpaC was developed recently, with significant antibody responses in mice and cross-protective immunity against both *S. dysenteriae* 1 and *S. flexneri* 2a [140].

#### 4.5.2. Other Subunit Vaccine Targets

Various subunit vaccine targets have emerged for combatting shigellosis, including outer membrane protein A (OmpA), known for their conservation and immunodominance among *Shigella* spp. [141]. OmpA is an outer membrane protein, which consists of two domains: an N-terminal β-barrel domain and a C-terminal globular domain [142]. OmpA is required for IcsA presentation, cell-to-cell spread, protrusion formation, and membrane integrity [143]. Recent advances include modifying the outer membrane vesicles from *S. flexneri* by combining a Tol-Pal system mutation with heat inactivation (HT-ΔtolR) to enhance outer membrane factor quality, resulting in a better immune response against *Shigella* and decreased toxicity [144]. Pan-*Shigella* surface protein 1 (PSSP-1), derived from the C-terminal polypeptide of IcsP, serves as another vaccine target. This conserved antigen is shared among different *Shigella* serotypes in Asia. Intranasal administration of PSSP-1 induces cross-protection against multiple *Shigella* serotypes, stimulating increased production of IL-17 and IFN-γ, along with enhanced antibody responses [145]. Furthermore, strategic interruption of the O-antigen polymerase gene (Δ*wzy*) was used to construct an attenuated *S. flexneri* 2a LAV strain, enhancing the exposure of surface proteins such as PSSP-1, IpaB, and IpaC, resulting in cross-protection against multiple *Shigella* serotypes in mouse models [146].

Additionally, new technologies enhance the design of vaccines against *Shigella* by selecting epitopes and peptides derived from subunit proteins. For example, reverse vaccinology has facilitated epitope mapping, identifying conserved virulence proteins in *S. sonnei*, thereby facilitating the screening of antigenic epitopes on their surface [147]. Such bioinformatics tools were used to determine potential vaccine candidates such as SigA [148], Pic, and Sap [149], for designing subunit vaccines containing B and T cell epitopes. Moreover, oral administration of peptides derived from the 49.8 kDa pili protein subunit of *S. flexneri* has been found to elicit robust intestinal immune responses in mice, including the activation of Th2 and Th17 cells, indicating it has potential for protecting against *S. flexneri* infection [150]. Q-S and D-K peptides from 49.8 kDa pili protein subunit have been shown to have a hemagglutinin activity and can mediate erythrocyte binding [131]. This 49.8 kDa pili protein has been shown to elicit significant increases in IL-17A levels, Th17 cells, and mucosal immune responses, suggesting they have potential as candidates for preventing *Shigella* and *Vibrio cholerae* infections [151].

Overall, subunit vaccines offer distinct advantages over live attenuated and inactivated vaccines. However, challenges such as cost implications, especially for developing countries targeted for future *Shigella* vaccines, and the requirement for specific adjuvants to enhance immune responses must be addressed as the field progresses.

## 5. The Important Role of Animal Models in Assessing *Shigella* Vaccines

To comprehensively study infectious organisms, establishing a natural symptomatic host model is crucial. Ideally, such a model should be both easily executable and replicative in nature. *Shigella*, a human-adapted pathogen, however, lacks a suitable small animal natural host model [152]. Although mice, guinea pigs, and rabbits can be susceptible to *Shigella* infection at non-gastrointestinal sites (e.g., lung infection or ocular infection), they are not amenable to simple oral challenge with dysentery-like symptoms [153]. Several animal models have been developed to overcome these challenges. These models range from surgical and other complex methods to more straightforward and easily executable approaches (Figure 4).

### 5.1. Mouse Pulmonary Model

The mouse pulmonary model serves as an innovative and efficient alternative for studying *Shigella* infections by circumventing challenges associated with oral challenge methods. In this model, mice are exposed to *Shigella* spp. through intranasal administration, effectively bypassing issues related to gut immunity and invasive procedures [154]. Upon *S. flexneri* exposure in the mouse pulmonary model, innate immune responses are triggered, such as the activation of TNF-α, infiltration of monocytes, and the presence of intra-cellular bacteria in bronchial and alveolar epithelia. While advantageous in inducing pneumonia and having a pathology that resembles human colitis, this model is limited by severe clinical scores, eventual death, and the absence of gut involvement, restricting its broader applicability. Nevertheless, the simplicity of this model still makes it a valuable tool for investigating the dynamics of *Shigella* infections [155].

### 5.2. Mouse Intraperitoneal Model

The mouse intraperitoneal model is a widely employed approach for studying *Shigella* infections in murine subjects in which *Shigella* is introduced into the peritoneal cavity through intraperitoneal injection (i.p.). Despite deviating from the natural route of infection, this model effectively reproduces key symptoms observed in human *Shigella* infections, such as severe diarrhea and weight loss [154]. This controlled experimental setting contributes significantly to our understanding of *Shigella* pathophysiology and provides some insight into the development of vaccines and treatments for *Shigella*-related diseases [78]. However, the intraperitoneal model of *Shigella* infection in mice poses significant drawbacks. It can lead to systemic pathogenesis instead of localized intestinal infection, resulting in rapid death from hepatitis or bacteremia. Moreover, this model is technically challenging and associated with suboptimal animal welfare conditions. It also fails to faithfully replicate the clinical manifestations of human shigellosis.

### 5.3. NAIP-NLCR4-Deficient Oral Mouse Model

The NAIP-NLCR4-deficient oral mouse model involves mice with deficiencies in NAIP (NLR family apoptosis inhibitory protein) and NLCR4 (NOD-like receptor family CARD domain-containing protein 4). In wild-type mice, the activation of the NAIP-NLCR4 complex protects the intestinal epithelium from *Shigella* invasion. In contrast, for NAIP-NLCR4-deficient mice, this protective mechanism is compromised, making them susceptible to *Shigella*-induced colitis [156]. This model provides new insight into the host–pathogen interactions during oral *Shigella* infection by shedding light on the role of the NAIP-NLCR4 inflammasome in defending the gastrointestinal tract against bacterial invasion. It has been particularly useful for understanding the molecular mechanisms underlying *Shigella* pathogenesis and the immune responses in the host following oral infection [157].

### 5.4. Zinc-Deficient Oral Mouse Model

The zinc-deficient oral mouse model has been used to explore the impact of zinc deficiency on the course of *Shigella* infection and the immune response in the host. Zinc is an essential micronutrient that plays a crucial role in various cellular processes, including immune function. This model has the advantage of allowing researchers to explore the molecular and cellular mechanisms through which zinc influences the host–pathogen interaction during *Shigella* infection. This contributes to a better understanding of the factors influencing the severity of *Shigella*-induced diseases [158]. However, the zinc deficiency model is unable to accurately replicate the intracellular location of *Shigella* in human infection, and it fails to replicate the complex relationship between malnutrition and the host inflammatory response [154].

### 5.5. Mouse Intracolonic Model

The mouse intracolonic model provides a method of studying *Shigella* infection by introducing the bacteria directly into the colon. This method involves the placement of a cannula into the colonic lumen, allowing controlled administration of the inoculum. Although this model offers advantages such as a lower required inoculum compared to other routes, it has limitations due to its surgical nature and potential deviations from natural infection pathways. Despite its effectiveness in inducing colonic changes resembling human colonic responses to *Shigella*, the invasive procedure and potential alterations in physiological conditions should be considered in its application and interpretation in research studies [159].

### 5.6. Mouse Xenotransplant Model

The mouse xenotransplant model is used to identify bacterial infection in a humanized context. This model involves transplanting human intestinal tissues into immunocompromised mice, creating an environment conducive to *Shigella* infection that closely mimics the human condition [160]. While this approach allows examination of the *Shigella* infection in human tissues and the human immune response, challenges include the need for immunodeficient mice and the potential influence of mouse-derived factors on the results. Careful consideration of these factors is essential to extract meaningful insights into *Shigella* pathogenesis and host interactions in this humanized context [161].

### 5.7. Guinea Pig Rectocolitis Model

The guinea pig rectocolitis model has also been used to study *Shigella* infections, particularly to study colonic pathogenesis. In this model, guinea pigs are used to investigate the effects of *Shigella* on the rectum and colon. The infection is induced through the intra-rectal administration of *S. flexneri*, allowing researchers to observe and analyze the resulting rectocolitis. One of the advantages of this model is its ability to replicate certain aspects of human *Shigella* infections, including the development of symptoms such as diarrhea, mucosal inflammation, and histopathological changes in the colon. The model provides insights into the local effects of *Shigella* on the colonic tissue, making it relevant for studying the pathophysiology of shigellosis. However, a limitation of the guinea pig rectocolitis model is that the high bacterial concentrations required for initiation of infection do not fully reflect the natural infection in humans. Additionally, the model primarily focuses on the colonic response, and researchers should be cautious when generalizing findings to other aspects of *Shigella* infections, considering that *Shigella* can affect different regions of the gastrointestinal tract [162,163].

### 5.8. Guinea Pig Keratoconjunctivitis Model

The guinea pig keratoconjunctivitis model is a valuable tool in *Shigella* studies, particularly for understanding the pathogenesis and effects of *Shigella* infections on the eyes. In this model, a specific dose, often around 5 × 10^8^ CFU of bacteria, is used to infect the conjunctival sac of one eye of the guinea pig, while the other eye remains uninfected as a control [164]. This acute infection induces keratoconjunctivitis, characterized by symptoms such as redness, swelling, and discharge from the infected eye. The infection typically resolves within a few days, especially with the administration of antibiotics. This model is particularly helpful in determining the correlation between *Shigella*-specific anti-LPS ocular sIgA and protection [163]. Overall, the guinea pig keratoconjunctivitis model provides insight into the ocular aspects of *Shigella* pathogenesis and the host’s immune response. However, this model is specific to studying the effects of *Shigella* on the eyes, and the broader systemic responses and complications that can occur in humans during *Shigella* infections, such as severe gastrointestinal symptoms, may be missed.

### 5.9. Rabbit Colonic Infection Model

The rabbit colonic infection model is used to investigate the pathogenesis and effects of *Shigella* infection on the colon. This model involves two primary methods: cecal bypass and colonic intubation [153]. The cecal bypass method entails placing a ligature near the ceco–colic junction, allowing ileo–ceco–colic communication while blocking the cecal luminal matters from entering the proximal colon. A cannula is then inserted approximately 10 cm distal to the ceco–colic junction, facilitating the introduction of an inoculum into the colon. This method, while effective, is considered tedious and time-consuming due to its surgical nature. This model allows for the study of acute infection, with observations including symptoms of acute discomfort, mucus–blood-containing stool, increased body temperature, tenesmus, internal hemorrhage, and signs of inflammation. Histological analysis reveals characteristic *Shigella*-mediated changes in the distal colon, resembling human colonic alterations post-*Shigella* infection. Despite its effectiveness, the model has limitations, such as the need for a high concentration of bacteria to initiate changes in the colon, making it less than ideal for certain studies.

### 5.10. Cynomolgus Monkey Model

The cynomolgus monkey model is good for understanding the pathogenesis and immune responses associated with intestinal *Shigella* infections. Macaques are susceptible to *Shigella*, and they can be challenged by either the intraduodenal route or the orogastric route [152]. In intraduodenal challenges, macaques are anesthetized, and a gastroscope is placed to reach the pylorus, followed by the introduction of the inoculum. The model provides insights into the host–pathogen interaction, as intraduodenal challenges induce the activation of *Shigella*-specific immune responses. While intraduodenal challenges may not exhibit overt clinical signs, the presence of *Shigella* can be confirmed through fecal plating, lasting for a specific duration post-challenge. On the other hand, orogastric challenges can give rise to clinical signs of disease, including the presence of anti-LPS antibodies. Despite providing valuable information about *Shigella* infection in primates, it is important to note that the dosage used in these models is much higher than the physiological dose encountered by humans. Ethical considerations and the refinement of experimental designs are crucial when utilizing non-human primate models in *Shigella* research.

### 5.11. Aotus Nancymaae Model

The *Aotus nancymaae* model is another non-human primate model for studying *Shigella* [165]. *Aotus nancymaae*, a species of owl monkey, has been used to investigate *S. flexneri* 2a 2457T infection. In this model, animals are deprived of food overnight, followed by neutralization of stomach acid using bicarbonate prior to bacterial exposure. The disease manifestation in the *Aotus nancymaae* model includes symptoms such as loose and bloody stools, reflecting characteristics observed in human *Shigella* infection. Post-challenge necropsy reveals pathological changes in the small intestine and stomach, with severely ill animals exhibiting necrotic tissues. This model provides an opportunity to study the pathogenicity of *Shigella* in a primate species, offering insights into disease mechanisms, immune responses, and potential interventions. However, like other animal models, it is essential to consider ethical aspects and animal welfare in the use of non-human primates for research purposes. The use of non-human primates can also be prohibitively expensive.

### 5.12. Controlled Human Infection Model (CHIM)

The Controlled Human Infection Model (CHIM) is a valuable tool in *Shigella* studies, providing a controlled environment for investigating human responses to *Shigella* infection. In this model, volunteers are deliberately exposed to a specific strain of *Shigella* under carefully monitored conditions. The initial cohort receives a relatively low infection dose, and subsequent cohorts may receive higher doses based on the observed attack rate. If the attack rate (AR) is less than or equal to 60%, the next two to four cohorts would receive a higher dose, while a lower dose would be used in case of an initial AR of more than 60% [166]. The final dose for the confirmatory cohort is selected based on the results of the previous cohorts. Upon receiving the challenge dose, subjects are monitored daily, with all stools collected for scoring and visual checks. Stools with a score of 3 to 5 are further assessed, and blood is confirmed by guaiac test. Anti-LPS IgG and IgA are monitored along with stool cultures. The CHIM allows researchers to evaluate the effectiveness of potential vaccines or treatments in a controlled setting, offering insights into immune responses, pathogenesis, and potential public health interventions. While this model provides a unique opportunity to study *Shigella* infection in humans, ethical considerations and stringent safety measures are paramount to ensure the well-being of study participants.

## 6. The Future of Developing a *Shigella* Vaccine

The urgency of developing a *Shigella* vaccine is highlighted by the escalating threat of *Shigella* infections exacerbated by climate change [167]. The changing climate, characterized by increased temperatures and altered precipitation patterns, expands environments conducive to *Shigella* proliferation and transmission [168]. Subunit vaccines, focusing on specific antigens associated with *Shigella* pathogenesis, emerge as a promising and safer alternative to LAV strains. By targeting key virulence factors, particularly those integral to the type III secretion system (T3SS), subunit vaccines can elicit targeted immune responses without the risks associated with LAVs, addressing the challenge posed by diverse *Shigella* serotypes.

The development of *Shigella* vaccines has witnessed significant advances in technologies aimed at enhancing efficacy, safety, and production ease. The integration of genomics, synthetic biology, and novel vaccine platforms has opened new avenues for effective and versatile *Shigella* vaccines. Bioinformatic tools allow researchers to sift through vast amounts of genomic data, identifying conserved proteins among *Shigella* strains with the potential to stimulate specific immune responses [149,169]. Furthermore, synthetic and recombinant biology methods are facilitating the design and construction of conjugate and chimeric proteins that incorporate and/or combine immunogenic regions from different *Shigella* antigens [11,170]. Additionally, self-adjuvant vaccine platforms and formulations represent another frontier in *Shigella* vaccine development by incorporating components that stimulate the immune system and eliminating the need for additional adjuvants [131,140]. These technologies hold promise in addressing challenges associated with *Shigella* infections, advancing vaccine candidates toward clinical applications, and improving overall safety profiles.

The future of developing a *Shigella* vaccine requires a comprehensive approach considering technological advancements and global challenges, such as antimicrobial resistance [171]. The rise of antibiotic-resistant *Shigella* strains poses a significant threat, emphasizing the importance of preventive measures through vaccination. Continuous research efforts are essential to stay ahead of emerging strains, considering the adaptability and evolutionary potential of *Shigella*. Collaborative initiatives between researchers, public health agencies, and pharmaceutical entities are crucial for expediting vaccine development and ensuring widespread accessibility, particularly in regions most vulnerable to *Shigella* outbreaks. Efforts should also be directed toward establishing effective vaccination strategies, considering age groups, dosages, and deployment in the context of evolving epidemiological patterns. Addressing these multifaceted challenges holds the potential to significantly impact global public health and reduce the burden of *Shigella*-related illnesses [172].

## Figures and Tables

**Figure 1 ijms-25-04329-f001:**
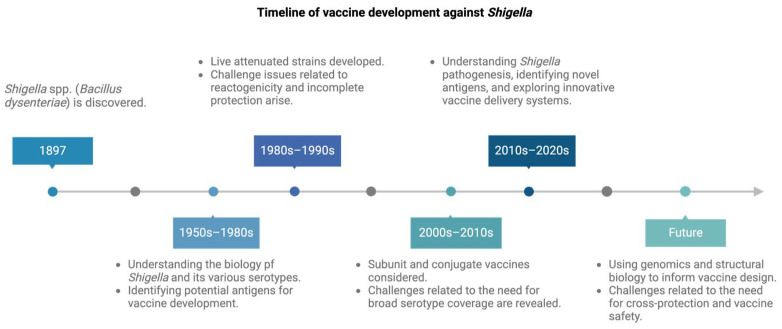
Timeline of vaccine development against *Shigella* infection. This figure illustrates key milestones in the field of *Shigella* research and vaccine development, highlighting breakthroughs and advances achieved over the years. Figure created using BioRender.com (accessed on 17 October 2023).

**Figure 2 ijms-25-04329-f002:**
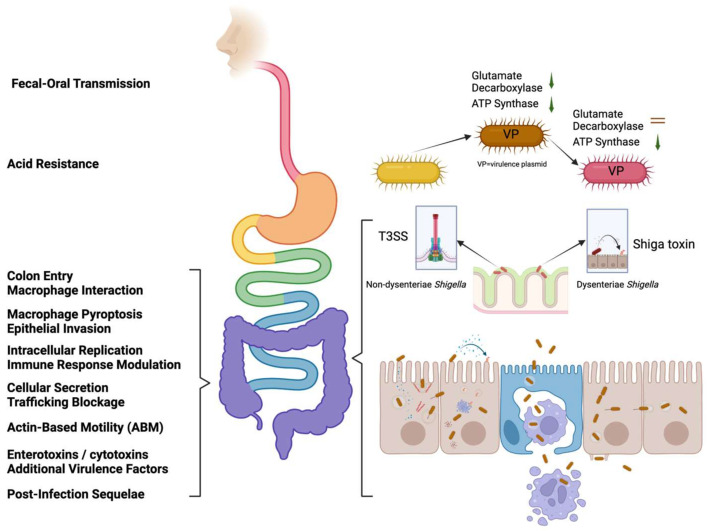
*Shigella* pathogenesis. (**Left**) Stepwise evasion of the host immune system by *Shigella* during its journey from ingestion to the colonization of the intestine. This figure highlights key mechanisms employed by *Shigella* to evade immune detection and clearance at various stages of infection. (**Right**) Detailed depiction of the process of *Shigella* invasion into host cells via genetic changes and acquisition of virulence plasmids (**right top**) providing insights into the intricate mechanisms by which *Shigella* breaches the host epithelial barrier and initiates intracellular replication by using the type III secretion system (T3SS) or enterotoxins (**right middle**), leading to infection and pathogenesis (**right bottom**). The straight arrows at the top represent mechanisms, while the straight arrows in the middle indicate different infections. Curved arrows in the middle and bottom denote the release of toxins. Figure created using BioRender.com (accessed on 17 October 2023).

**Figure 3 ijms-25-04329-f003:**
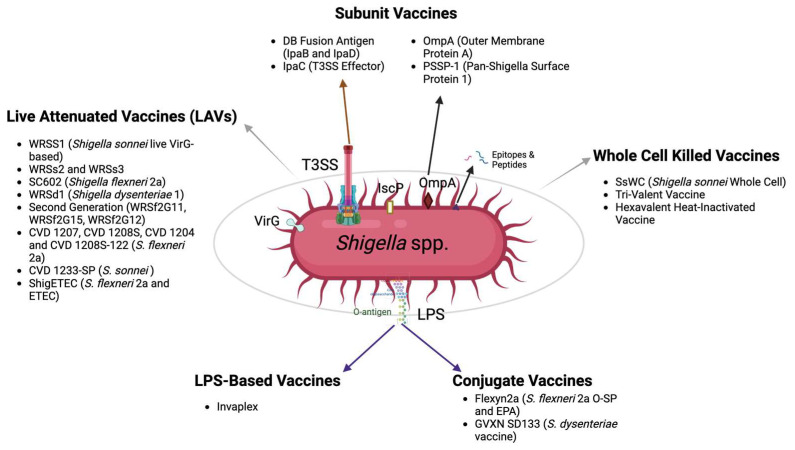
Current *Shigella* vaccines in development. This figure provides an overview of the diverse vaccine candidates currently under development against *Shigella* infections. Each vaccine candidate is displayed along with its corresponding target antigens. Figure created with BioRender.com (accessed on 17 October 2023).

**Figure 4 ijms-25-04329-f004:**
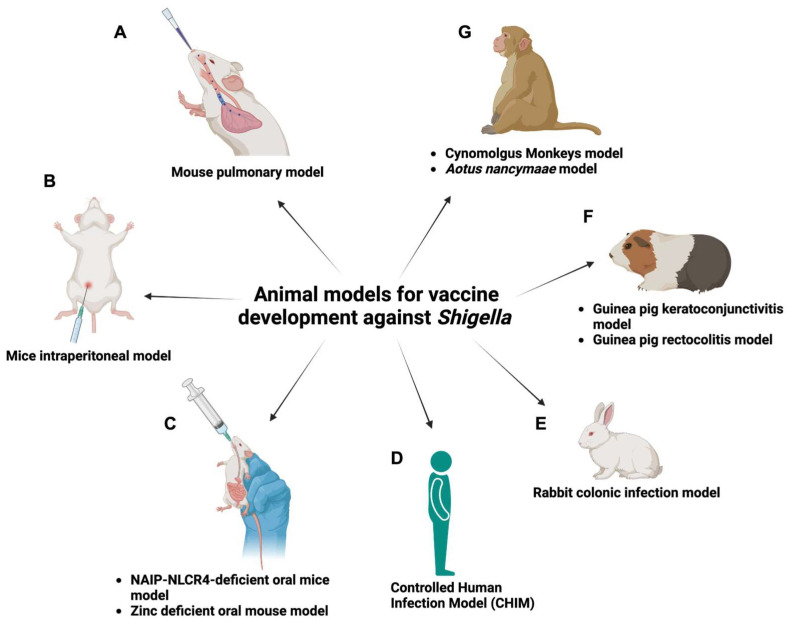
Animal models for vaccine development against *Shigella* infection. (**Left**) Various challenge routes can be used in mouse models. The figure outlines different routes, such as (**A**) intranasal delivery, (**B**) intraperitoneal injection, and (**C**) oral administration. (**Right**) Non-mouse animal models employed in *Shigella* vaccine development research. This includes (**D**,**G**) primate models, (**E**) rabbit models, and (**F**) guinea pig models. Each offers unique advantages and insights into *Shigella* pathogenesis and vaccine responses. Figure created using BioRender.com (accessed on 17 October 2023).

**Table 1 ijms-25-04329-t001:** Respective stages of development of the different vaccine candidates against *Shigella* spp.

Vaccine Type	Development Stage
Whole-cell formalin-inactivated vaccine (SsWC)	Phase 1 human trial
Trivalent vaccine using whole-killed *S. flexneri* 2a, *S. sonnei*, and *S. flexneri* 3a	Preclinical
Hexavalent heat-inactivated vaccine covering multiple *Shigella* strains	Preclinical
*S. sonnei* live virG knockout vaccine candidate, WRSs1	Phase 1 human trials completed
*S. sonnei* live attenuated vaccine candidates, WRSs2 and WRSs3	Phase 1 trials completed
*S. flexneri* 2a SC602	Phase 1 trials ongoing
*S. dysenteriae* 1 SC599	Phase 1 trials completed
*S. dysenteriae* 1 live attenuated vaccine candidate, WRSd1	Preclinical
Second-generation live attenuated vaccine candidates (e.g., WRSf2G11, WRSf2G12, WRSf2G15)	Preclinical
*S. flexneri* 2a live attenuated vaccine candidate, CVD 1207	Phase 1 trials ongoing
*S. flexneri* CVD 1208S and CVD 1204	Phase 1 trials ongoing
Combined *Shigella*-ETEC vaccine candidate, CVD 1208S-122	Preclinical
Live attenuated *S. sonnei* vaccine strain, CVD 1233-SP	Preclinical
ShigETEC (live attenuated combined vaccine targeting *S. flexneri* 2a and ETEC)	Phase 1 clinical trials ongoing
*Salmonella* vaccine system for delivering *S. flexneri* 2a (Sf2a) O-antigen	Preclinical
RASV-delivered Sf2a O-antigen vaccine	Preclinical
Oral *Shigella* 2aT32-based vaccine expressing a fusion antigen of UreB-HspA	Preclinical
nvaplex (*Shigella* LPS combined with T3SS proteins IpaB and IpaC)	Phase 1 studies completed
Invaplex AR (second-generation artificial Invaplex product)	Phase 1 studies completed
Ac3-S-LPS (tri-acylated lipid A variant)	Phase 1 studies completed
PLVF (pentavalent LPS candidate vaccine)	Preclinical
Needle-free delivery systems for OMVs derived from *S. flexneri*	Preclinical
Flexyn2a (*S. flexneri* 2a O-SP conjugated with EPA)	Phase 1 studies completed
Sd1-EPA and Sf2a-EPA (bioconjugates)	Phase 1 studies ongoing
GVXN SD133 (*S. dysenteriae* vaccine featuring *Shigella* O1 LPS and EPA)	Phase 1 studies completed
*Shigella* 4V (quadrivalent bioconjugate vaccine)	Preclinical
SCV (multivalent *Shigella* conjugate vaccine)	Preclinical
*Shigella* O-polysaccharide (OPS)-IpaB conjugate vaccine	Preclinical
Various subunit vaccines targeting T3SS proteins (e.g., IpaB, IpaD, IpaC)	Varying stages of preclinical and clinical development
Subunit vaccines targeting other *Shigella* proteins (e.g., OmpA, PSSP-1)	Varying stages of preclinical and clinical development
Peptide-based vaccines derived from subunit proteins (e.g., peptides from 49.8 kDa pili protein subunit)	Preclinical

## Data Availability

Not applicable.

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
