# Peer review of "Shigella Vaccines: The Continuing Unmet Challenge"

_ijms, 2024, doi:10.3390/ijms25084329_

Round 1

Reviewer 1 Report

Comments and Suggestions for Authors

In this paper, the authors aim at providing a thorough overview and evaluation of the current progress of Shigella vaccine candidates that are under development.

The manuscipt is an interesting narrative review of the available literature, that provides valuable insights and inputs for further research.

My only suggestoin would be to reduce the length of the article, as it is quite long, unless the eidtors agree to the current status.

Author Response

We understand your concern, and we acknowledge that balancing the level of detail in the manuscript is crucial. However, we also want to be sure to provide comprehensive coverage of the topic, addressing the requirements and expectations of all reviewers. As such, we have carefully considered the suggestions and requests from each reviewer and have modified our manuscript accordingly to incorporate the necessary detail and information. We hope that you will find the updated content beneficial and we feel that it has contributed positively to the overall quality of the manuscript.

Reviewer 2 Report

Comments and Suggestions for Authors

The review manuscript entitled: "Shigella vaccines: the continuing unmet challenge"  by Ti et al. The authors explored Shigella pathology, with a primary focus on the status of Shigella vaccine candidates. The review manuscript highlihts recent and future advances in the field, however it lacks a detail understanding of existing genes and possibile targets for novel theraputic approachses.

Major comments:

-In the abstract authors mentioned potential Shigella immunogens, such as conserved T3SS proteins are there some other such targets from the literature. Can authors make over of their structures.

-Authors discused antibody theraputic approach applied, for example: researchers found that vaccination elicited functional antibody responses and induced Th1/Th17 signature cytokines in Bangladeshi adults and children for WRSs1. Can they list the sequence of structural relationship between different antibodies.

-More details about drug option for fluoroquinolone-resistant Shigella infections, involving ceftriaxone, pivmecillinum, and azithromycin can be included. Chemical structures target genes etc.

-All figure captions required detail description of the figures. I hardly find any information in captions.

Comments on the Quality of English Language

Some sentences are difficult to understand, maybe they can be simplified.

Author Response

Major comments:

“In the abstract authors mentioned potential Shigella immunogens, such as conserved T3SS proteins are there some other such targets from the literature. Can authors make over of their structures.”

We have expanded the abstract to incorporate other relevant immunogens identified in the literature. Additionally, we have included detailed structural information for each of these targets in the main manuscript, ensuring comprehensive coverage of potential Shigella immunogens used for vaccine development. (Line 18, 172-181, 184-187, 539-542)

“Authors discused antibody theraputic approach applied, for example: researchers found that vaccination elicited functional antibody responses and induced Th1/Th17 signature cytokines in Bangladeshi adults and children for WRSs1. Can they list the sequence of structural relationship between different antibodies.”

We have included a comparison between the structural characteristics of different antibodies identified in the mentioned publication. This comparison highlights the sequence of structural relationships between these antibodies and provides insight into their potential therapeutic applications. (Line 326-330)

“More details about drug option for fluoroquinolone-resistant Shigella infections, involving ceftriaxone, pivmecillinum, and azithromycin can be included. Chemical structures target genes etc.”

We have carefully addressed your comment by incorporating additional details about the drug options for fluoroquinolone-resistant Shigella infections, specifically ceftriaxone, pivmecillinam, and azithromycin, into the manuscript. We have provided comprehensive information, including naming the chemical structures of antibiotic classes and their target genes, to enhance the understanding of these antibiotics and their efficacy against fluoroquinolone-resistant Shigella strains. (Line 221-230)

“All figure captions required detail description of the figures. I hardly find any information in captions.”

We have expanded each caption so that they collectively provide comprehensive information about the content and purpose of the corresponding figure.  This should allow readers to understand the significance of the visual representation provided. (Line 39-41, 197-201, 285-287, 583-588)

Reviewer 3 Report

Comments and Suggestions for Authors

Very nice and comprehensive review. Describes all the details for Shigella vaccines. Well organized into sections with mjor focus on different types of vaccines.

Just one suggestion (not mandatory). A table of vaccines, categorized into different types and listed as stage of development (preclinical, or clinical Phase 1-4) will be a good summary of existing literature.

Just one Typo on Line 232: change "while" to "whole"

Author Response

While we appreciate the value that such a table could bring to the manuscript, we have carefully considered the length of the paper and the time constraints involved in creating and incorporating a new table at this stage of the revision process. As such, we have decided not to include the suggested table in the revised manuscript. The typo you pointed out on line 232 has already been corrected during the language modification process.

Round 2

Reviewer 2 Report

Comments and Suggestions for Authors

Authors tried their best to consider my comments, however, I don't see any structure comparision of protien or antibodies where i comment in my previous review.

Comments on the Quality of English Language

NA

Author Response

We do not understand the comment pertaining to antibody therapeutic approach. We do no discuss such therapeutics. The target audience for a Shigella vaccine cannot afford such an approach. The sentence discussed that vaccines producing functional antibodies and cytokines that are required for protection. The majority of the proteins that we specifically discuss still do not have a solved structures. 

Additionally, we made some edits in the manuscript to make several of the longer sentences shorter. Further editing is not warranted as both of the other two reviewers had no issues with the English giving us 5 stars. 

We hope this clears up the questions of reviewer 2